# The Ecotoxicology of Microbial Insecticides and Their Toxins in Genetically Modified Crops: An Overview

**DOI:** 10.3390/ijerph192416495

**Published:** 2022-12-08

**Authors:** Eduardo C. Oliveira-Filho, Cesar K. Grisolia

**Affiliations:** 1Embrapa Cerrados, Rodovia BR020, Km 18, Planaltina, Brasília 73310-970, DF, Brazil; 2Departamento de Genética e Morfologia, Instituto de Ciências Biológicas, Universidade de Brasília, Brasília 70910-900, DF, Brazil

**Keywords:** biopesticides, ecotoxicity, ecosystems, biological insecticides, transgenic plants, *Bacillus thuringiensis*

## Abstract

The use of microbial insecticides and their toxins in biological control and transgenic plants has increased their presence in the environment. Although they are natural products, the main concerns are related to the potential impacts on the environment and human health. Several assays have been performed worldwide to investigate the toxicity or adverse effects of these microbial products or their individual toxins. This overview examines the published data concerning the knowledge obtained about the ecotoxicity and environmental risks of these natural pesticides. The data presented show that many results are difficult to compare due to the diversity of measurement units used in the different research data. Even so, the products and toxins tested present low toxicity and low risk when compared to the concentrations used for pesticide purposes. Complementary studies should be carried out to assess possible effects on human health.

## 1. Introduction

Microbial pesticides have been shown, over the last twenty years, to be a promising alternative to chemical pesticides, both in agriculture and mosquito control. Using microbial pest control agents (MPCAs) or entomopathogens as biological insecticides is a viable and interesting alternative because the selected and used microorganisms are naturally found in the environment [1]. Regardless of their origin, however, the fact that they have a biocidal effect raises concerns about possible impacts on the environment and the health of the exposed population. The key question is if the application of a natural product is in a non-natural way or if the quantity is safe for environmental and human health. Furthermore, due to the importance and widespread use of microorganisms in biological control, utilizing toxins cloned in the main cultivated crops, these concerns have become particularly important for ecotoxicological research.

Ecotoxicology is the science that studies the fate and effects of chemicals on living organisms. If the main mechanisms of action of some microbial insecticides have the same effects as chemical toxins, then this can be considered a toxicological effect. The hazard potential is not based only on the toxic effects of compounds but also on their uptake and elimination kinetics, their bioavailability, dispersion, or accumulation in the environment [2]. 

All over the world, pesticides, be they biological or chemical, must be registered with government agencies. During this process, several toxicological and ecotoxicological assays are performed to evaluate the human and environmental safety of the future end-use product. These studies include toxicological and ecotoxicological assays with mammals and other aquatic and terrestrial organisms that predict environmental hazards. Indeed, ecotoxicological assays are increasingly important in establishing the effects on living organisms of hazardous materials, including biological pesticides, due mainly to the potential risk of transfer to different ecological niches. They are also vital in assessing environmental quality to maintain healthy natural life.

In this context, the United States Environmental Protection Agency (USEPA) has developed several guidelines for evaluating ecotoxicity caused by biopesticides to non-target organisms, including aquatic and terrestrial species [3,4]. The protocols do not differ much from the classic toxicological tests since the main objective is to determine the presence of any hazardous toxin in the strains of microorganisms and estimate the effects of these strains and their toxins on non-target species beyond the potential for pathogenicity [5]. By definition, microbial insecticides are also called entomopathogenic insecticides or insect pathogens, and, in general, these bioproducts have specific mechanisms of action, including their toxins, and are quite harmless to non-target species [6]. 

It should be noted, however, that products such as genes or metabolites from these insect pathogens have been used to prevent crop damage using transgenic strategies that allow the transference of genes from one species to another. Then the microbial toxins can be expressed in genetically modified crops [7]. However, as defined by toxicology, all chemicals are toxic. Therefore this overview aims to present the ecotoxicity of the main microbial insecticides and their potential adverse effects on non-target species. Studies that examine the ecotoxicity of endotoxins in genetically modified crops are included, all based on an overview of the published data.

## 2. Microbial Pesticides

### 2.1. Bacteria-Based Pesticides

As it was one of the first microorganisms to be used as an insecticide, *Bacillus thuringiensis* (Bt) is one of the most studied bacteria. Although in several countries the current phase system is the main strategy for establishing ratings for biopesticides as an environmental hazard, various other investigations have been conducted with Bt. 

This entomopathogenic bacterium has a highly specific mechanism of action since, during the sporulation cycle, it forms a protein crystal that becomes toxic when ingested by susceptible insects. This toxin binds to receptors in the midgut epithelium, causing cell lysis and insect death [8].

To assess the potential impacts of *Bacillus thuringiensis* on aquatic invertebrates, several trials with different species were carried out. Among these, *Daphnia magna*, *Cyclops* sp., and *Rivulogammarus pulex* were not affected by the biopesticide. Still, the crustacean *Chirocephalus grubei* of the Anostraca order showed a mortality of 57% when exposed to a concentration of 18 ppm, equivalent to 100 times the concentration of the larvicide used for mosquito control [9]. In another study, the authors did not observe concentration/response effects, in concentrations up to 1.5 × 10^6^ CFU/mL, on the aquatic crustacean *Daphnia similis* [10]. In a more complex approach, Chen et al. [11] did not observe significant effects on the development, reproduction, and reproductive parameters of the crustacean *Daphnia magna* after exposure for 21 days to a concentration of 500 µg/L of purified Cry1C protein.

In studies with mollusks, flatworms, and amphibians, no adverse effects were observed after exposure to a concentration of 180 ppm [12]. In another piece of research conducted in the United States, Merritt et al. [13] reported no evidence of effects on the community of aquatic invertebrates after applying a field control program. In a laboratory study using the aquatic snail *Biomphalaria glabrata*, Oliveira-Filho et al. [10] found a 30-day LC50 of 1.5 × 10^7^ CFU/mL. 

In a study with some fish species exposed for 30 days at concentrations of between 10^9^ and 10^10^ colony-forming units (CFU)/mL, there was no evidence of mortality, pathogenicity, or infectivity. In another piece of research with *B. thuringiensis kurstaki*, 20% mortality was observed in trout exposed for the 32 days of the experiment, and this mortality was attributable to excessive competition for food in the water, greatly blurred by the presence of high concentrations of the microorganism in suspension [14]. Mittal et al. [15] fed larvae of the fish *Poecilia reticulata* contaminated with various chemical and biological insecticides, and there was no mortality in these larvae when fed with Bt.

Following the same line of thought, Grisolia et al. [16] observed no mortality in the fish species *Danio rerio* and *Oreochromis niloticus* when exposed for 30 days at a concentration of 5 × 10^6^ CFU/mL.

However, Snarski [17] showed mortality of the fish *Pimephales promelas* in the larval stage exposed to concentrations of *Bacillus thuringiensis israelensis* (BTI) of the order of 10^6^ CFU/mL.

Concerning effects on soil invertebrates, Addison [18] observed that nematodes and beetles may be at risk after the application of Bt. According to the author, all Bt strains and isolated toxins tested were toxic to eggs of the nematode *Trichostrongylus colubriformis* in concentrations of 0.001 to 130 µg/mL. On the other hand, in a study with the beetle *Digitonthophagus gazela* [10], the authors obtained a calculated LC50 of 1.3 × 10^6^ CFU/g after 30 days of exposure to Bt.

The acute toxicity and pathogenicity of different Bt commercial formulations were evaluated using several species of birds, including *Colinus virginianus*, a quail species, and *Anas platyrhynchus*, a duck species, by administering them orally in doses of 10^9^–10^11^ CFU/kg/day. The species tested presented no adverse effects during the observation period [19].

Innes and Bendell [20] evaluated the effects of a commercial formulation of *B. thuringiensis kurstaki* on populations of small mammals for 90 days. The observed results suggested that ingesting contaminated insects did not cause adverse effects for these populations.

In fact, the problem of insecticides based on *B. thuringiensis* has been its effect against non-target insects [21]. Data published show that 10 orders of insects are susceptible or may suffer some damage after exposure to *Bacillus thuringiensis*. Of these, the order Lepidoptera is the most affected, with 572 susceptible species, followed by Diptera with 266 species, Coleoptera with 106, Hymenoptera with 62, Hemiptera with 48, Syphonaptera with 7, Orthoptera with 6, Isoptera with 5, Neuroptera with 4 and Thysanoptera with 3, totalizing 1079 species [22].

*Lysinibacillus (Bacillus) sphaericus* is a mosquito entomopathogen but has a markedly smaller range of target mosquitoes than Bt. As well as Bt, ingested toxins are solubilized in the alkaline midgut and cleaved by proteases [23].

According to Lacey and Siegel [19], compared to Bt, there are few data available on the ecotoxicity of *Bacillus sphaericus* to invertebrates. As this species is extremely efficient in controlling mosquito larvae, obtaining data on its adverse effects on aquatic organisms is very important. In a study conducted in India [24], the effects were tested on two strains of *B. sphaericus* of the freshwater crustaceans, *Daphnia similis* and *Streptocephalus dichotomus*, as well as on the annelid *Tubifex tubifex*. For all tested species, the adverse effects became evident only at concentrations ranging from 2500 to 27,000 times greater than the one necessary to obtain the larvicidal effect. 

Another study with the fish *Procambarus clarkii* [25] observed lethal effects only at concentrations 1000-fold greater than that required to kill insect larvae. According to Walton and Mulla [26], in a mesocosm experiment using the fish *Gambusia affinis* and the bacterium *B. sphaericus* to control mosquito larvae, there was a sharp decrease in the number of larvae, but no adverse effects on fish were noted. Saik et al. [27] reported several studies with mammals with no significant results for adverse effects. According to the authors, several strains of *Bacillus sphaericus* were injected in different ways into mice, rats, guinea pigs, and rabbits for acute and chronic evaluations, and of the few lesions observed, several were also recorded in animals receiving autoclaved material, probably indicating physical damage.

In a more recent study, neither mortality nor visible adverse effects were reported in the fish species *Danio rerio* and *Oreochromis niloticus* exposed to concentrations of *B. sphaericus* of 5 × 10^6^ CFU/mL, suggesting that the tested strains have LC50s higher than the maximum concentration tested.

In another publication, *B. thuringiensis* and *B. sphaericus* in concentrations of 10^6^ spores/mL were not lethal to the fish *Hyphessobrycon eques*, also called Mato Grosso fish, in Brazil [28].

For the microcrustacean *Daphnia similis* [10], an EC50 was also observed for immobilization greater than 1.5 × 10^5^ spores/mL of *B. thuringiensis israelensis* (BTI). In the same study with *B. thuringiensis kurstaki* (BTK) and *Bacillus sphaericus* (BS), the EC50 was greater than 1.5 × 10^6^ spores/mL. For the aquatic snail *Biomphalaria glabrata*, the authors observed that, with BTK and BS of two different strains, concentrations up to 5 × 10^7^ did not cause any lethal effect, and the LC50s were considered higher than 5 × 10^7^. On the other hand, with BTI, the LC50 for *B. glabrata* after 30 days was determined to be 1.5 × 10^7^ spores/mL.

The non-target beetle *Digitonthophagus gazella* was not susceptible to concentrations of BTI or BS strains up to 1.75 × 10^6^ spores/mL, but for BTK, it was possible to calculate an LC50 of 1.3 × 10^6^ with confidence limits of 5.6 × 10^5^ to 2.8 × 10^6^ spores/mL [10].

As regards ecotoxicology, the fate of bacterial insecticides, particularly the fate of bacteria, is another point to evaluate. The circulation of bacteria and their capacity to colonize their host is an important question in evaluating the potential infectivity of a microbial pest control agent. The bacterial elimination or clearance has been investigated in some papers.

Snarski [18], studying the persistence of *B. thuringiensis israelensis* in the fish *Pimephales promelas*, reported that the spore count was rapidly reduced after transferring fish to clean water, with a decrease of around 3 orders of magnitude in 1 day. This author showed that spores were rarely detected in fish after 8 days (192 h). In a similar study [28], it was demonstrated that the Brazilian fish *Hyphessobrycon eques* and snail *B. glabrata* eliminated the spores of *B. thuringiensis* and *B. sphaericus* tested strains from their body after 7 days of recovery in clean water. In snails, it was observed that CFUs of *B. sphaericus* were found in higher numbers after 68 h of recovery, but in any case, the tendency presented by the obtained data shows that total elimination is imminent and that the persistence of spores or the bacteria for a time does not seem to be deleterious to the organism.

In general, most of the research that found some adverse effect claimed that the damage was justified by physical contact or by a high concentration of the product, which can lead to increased turbidity in the water, a decrease in dissolved oxygen, or possible toxicity of chemical components present in the formulation of the products. 

Recently, Swiss albino mice were exposed intraperitoneally to *B. thuringiensis* spore-crystals var. *kurstaki*, Cry1Aa, 1Ab, 1Ac, and 2Aa at concentrations of 27, 136, and 270 mg/kg for 24 and 72 h. Hematological disturbance occurred at the highest exposure levels. Cytotoxicity and genotoxicity were also evaluated, resulting in micronucleus induction and bone marrow cell inhibition at the highest exposure levels, which are not commonly found in the environment [29]. The same study protocol was carried out previously, but changing to oral exposures resulted in no genotoxicity and low cytotoxicity. Thus, it was demonstrated that oral exposure is less toxic than intraperitoneal exposure, showing the breakdown of these cry-endotoxins by the gastrointestinal tract [30].

Table 1 presents a summary of the main ecotoxicity data regarding bacterial-based pesticides. 

### 2.2. Fungi-Based Pesticides

Although most fungi used in biocontrol have insecticidal activity, some species are also used in agriculture against weeds or other fungi [31]. *Metarhizium anisopliae* is an entomopathogenic fungus that is present in soils throughout the world. It was first found to be a biocontrol agent in the 1880s. This fungus can control four types of insect pests (beetles, termites, spittlebugs, and locusts) [32]. The water fungal insecticide *Lagenidium giganteum* was also tested against several species in classical ecotoxicity tests. Survival and reproduction of the microcrustacean *Ceriodaphnia dubia* were affected in concentrations above 6250 spores/mL [33].

Adverse effects of the fungus *Trichoderma stromaticum* were evaluated on the freshwater microalgae *Selenastrum capricornutum*, the microcrustacean *Daphnia similis*, fish *Hyphessobrycon scholzei,* and Wistar rats representing mammals. The results showed no adverse effects for the algae, fish, and rats, but 30% of the *Daphnia similis* reproductive rate was inhibited at a concentration of 10^6^ spores/mL. The authors of that study reported that, due to this result, more studies of Phase II are needed to evaluate the persistence of the fungus in the aquatic environment [34].

The fungi *Metarhizium anisopliae* and *Beauveria bassiana* are registered for use in Brazil and other countries. Genthner and Middaugh [35] evaluated the effects of *M. anisopliae* on embryos of the inland silverside fish *Menidia beryllina*. In this study, several adverse effects on embryos and newly hatched larvae were found. Genthner et al. [36] also showed the effects of this fungus on embryos of the crustacean *Palaemonetes pugio* and the frog *Xenopus laevis*. Regarding *Beauveria bassiana*, the teratogenic effects of this fungus were also observed on embryos and larvae of inland silverside fish *M. beryllina* [37]. In another study [38], an LC50 was recorded with 0.56 mg/L of beauvericin, a toxin released by some strains of *B. bassiana* on the crustacean *Mysidopsis bahia*. The authors observed mortality of organisms in concentrations from 1.5 × 10^6^ spores/mL.

Recently Bordalo et al. [39] tested the lethality of a *Beauveria bassiana* commercial product and found a 48-h LC50 of 34.7 mg/L for the aquatic insect *Chironomus riparius*.

No adverse effects were obtained on nymphs of mayfly, *Ulmerophlebia* sp., exposed in laboratory tests to *M. anisopliae* var. *acridum* at the concentration of 2.0 × 10^6^ conidia/mL, while the same concentration caused 100% mortality in the cladoceran, *Ceriodaphnia dubia*, after an exposure of 48 h. At 6.7 × 10^3^ conidia/mL, there was only 5% mortality after 192 h [40].

Many studies dealing with the possible side effects of *M. anisopliae* on non-target organisms in the laboratory and in the field, especially on predators, parasitoids, honey bees, and earthworms, are presented in an extensive review [41].

The effects of a fungal suspension of 1 × 10^9^ spores of *M. anisopliae* were evaluated after being fed to the leopard frog, *Rana pipiens*. No mortality or recovery was recorded in any of the tissues. The viscera were free of fungal elements [42].

Avian safety studies were performed with the Japanese quail [43]. The test birds were allowed to consume spore suspensions of *M. anisopliae*. The total number of spores consumed was 4.9 × 10^10^/bird. There was no mortality or abnormal behavior in the experimental birds. Positive recoveries of *M. anisopliae* were carried out on plates streaked with fecal washings. *M. anisopliae* was recovered from heart and lung smears of two test birds. However, careful histological examination showed no evidence of spores or hyphae in these tissues.

The fungus *Sporothrix insectorum* was also tested and did not cause any acute effect on the aquatic snail *B. glabrata* or the beetle *D. gazella* at concentrations of 5.0 × 10^7^ and 1.25 × 10^7^ spores/mL, respectively, after 30 days of exposure [10]. In another study, Jonsson and Genthner [44] show an absence of effects of the fungi *Colletrotrichum gloeosporioides* on the crustaceans *Palaemonetes pugio* and *Artemia salina.*

In mammals, this fungus species was tested for toxicity and pathogenicity by the intramuscular route in mice without observation of infection, multiplication, or persistence of the microorganism after three days [45]. 

Table 2 presents a summary of the main ecotoxicity data regarding fungi-based pesticides.

### 2.3. Virus-Based Pesticides

A large number of viruses have the potential for insect biological control. Of these, the most studied are in the Baculoviridae group, of which the main genera are Nucleopolyhedrovirus (NPV) and Granulovirus (GV). Overall, these viruses have focused their action spectrum on about 43 species of 11 families of the lepidopteran order [46]. The primary viral transmission mechanism in Lepidoptera is through the release of particles called occlusion bodies or polyhedral inclusion bodies (PIBs).

Although viruses are among the least studied biological insecticides, this was one of the first groups to apply for registration in the United States, along with Bacillus. The *Heliothis* NPV was undoubtedly the most extensively studied among entomopathogenic viruses, not only on vertebrates but also on invertebrates and plants. Since 1963, the *Heliothis* NPV has been systematically tested against several non-target species to ensure the safety of its use. The dosages and concentrations used were 10 to 100 times higher than the average used in the field. No toxicity or pathogenicity was observed in any of the assays except for the target species [47]. The first viral insecticide Elcar™ was introduced by Sandoz Inc. in 1975. Elcar™ was a preparation of *Heliothis zea* NPV, which is a relatively broad-range baculovirus and infects many species belonging to the insect genera *Helicoverpa* and *Heliothis* [48]. HzSNPV provided control of cotton bollworms but also of pests belonging to these genera that attack soybean, sorghum, maize, tomato, and beans. In 1982, Sandoz decided to discontinue production [49].

The use of a baculovirus as a biopesticide has been successful in the case of *Anticarsia gemmatalis* nucleopolyhedrovirus (AgMNPV), employed to control the velvet bean caterpillar in soybean. This program began in Brazil in the early 1980s, and over 2000 ha of soybean were treated with the virus [50]. Table 3 presents the results of two Nucleopolyhedrovirus ecotoxicity studies used by the United States Environmental Protection Agency to renew the registration of these viral insecticides. In all cases, no adverse effects were observed in the non-target organisms tested [51].

The NPV of the red-headed pine sawfly (*Neodiprion lecontei*) was tested against the rainbow trout, *Salmo gairdneri*, and the microcrustacean *Daphnia pulex*, and no adverse effects were observed [56].

Kreutzweiser et al. [57] observed weight loss in rainbow trout (*Oncorhynchus mykiss*) fingerlings exposed to food contaminated with spruce budworm (Nucleopoliedrovirus *Choristoneura fumiferana*) in the order of 9.6 × 10^8^ occlusion bodies per fish larvae. The authors reported that this effect was observed in the control group, too, and stressed that it did not appear to be related to the treatment.

In a field study [58], a clear absence of adverse effects of *S. frugiperda* NPV on natural enemies or non-target insects was demonstrated.

Regarding *Cydia pomonella* GV, ecotoxicity data for fish, aquatic invertebrates, algae, and aquatic plants indicated low toxicity. A 14-day acute toxicity study to assess the effects on earthworms also showed no lethality [59].

## 3. Bt Genetically Modified Crops

Cry-proteins from *Bacillus thuringiensis* (Bt) are among the most cloned protoxins in genetically modified plants. Today, many transgenic plants based on cry-endotoxin genes and their corresponding food are available everywhere. Transgenic plants are cloned with these Bt genes expressing protoxins that kill the caterpillars, acting as a natural plant insecticide. Almost all genetically modified cotton, corn, and soybean crops code for Cry-endotoxins of *Bacillus thuringiensis*. Besides, bioinsecticides containing Bt-endotoxins are widely applied to control the *Aedes aegypti* mosquito in urban areas by government programs against dengue and other related diseases [60].

The Bt-corn endotoxin is degraded more rapidly in water than in soil and has shown to be resilient in both ecosystems [61]. A reassessment of Bt corn and Bt cotton carried out by the US-Environmental Protection Agency (USEPA) in 2001 concluded that they do not pose risks to the environment and human health. The US-Food Chemical Administration (USFDA) determined that transgenic crops with Bt-genes inserted are as safe as their conventional counterparts. The US FDA also states that companies producing new transgenic plant varieties with new toxins or a protein that could present as a new allergen require labeling to inform consumers of changes to the food. Because of this reassessment, the EPA approved registration renewal in the USA [62].

### 3.1. Risks to Soil Organisms

There are concerns about the environmental risks posed by these cry-endotoxins released through root exudates from transgenic crops. These toxins are strongly adsorbed by minerals in clay soils, enhancing their persistence and maintaining their insecticide properties. In contrast, regarding sandy soils, Bt-toxin levels decline more rapidly, mainly due to physico-chemical breakdown [63,64]. According to the USEPA reassessment, most of the Cry-endotoxins deposited in soil by Bt crops degraded rapidly (1–15 days), but residues persisted in their biologically active form for more than 40 days. Studies with soil bacteria, actinomyces, fungi, protozoa, nematodes, springtails, or earthworms revealed low toxicological risks [65]. However, the accumulation of Bt Cry-endotoxins in soils and the lack of toxicity to non-target organisms should be further clarified [62,65].

Flores et al. [66] showed that the breakdown of Bt plants in soil is slower than that of non-Bt plants. The lower biomass degradation from Bt plants is not directly related to soil microbiota. They observed these effects by comparing degradation rates between Bt corn and non-Bt corn. In this case, the lignin content of Bt corn is significantly higher than non-Bt corn. Furthermore, other Bt plants such as canola, cotton, potato, rice, and tobacco showed a non-significant but consistently higher content of lignin than their respective non-Bt. The authors suggested that the higher amounts of lignin could interfere with the rate of biodegradation of transgenic plant biomass.

Studies on the fate and behavior of Bt and Cry toxins in soils came up only after intensive spraying of Bt for the biological control of pests and the commercialization of genetically modified plants. The persistence of these cry toxins in soils was evaluated concerning microbiological activity, solubilization, soil pH, adsorption by soil organominerals, temperature, and UV radiation. Research has been carried out in the laboratory and field studies with pure Cry1Ac endotoxin and a commercial formulation of Bt biopesticide. Regarding the spraying of commercial formulations of Bt biopesticides, the half-life of detectable Cry1Ac toxins mainly depends on temperature and sunlight conditions, being determined between 1 day (25 °C) and two to four weeks (4 °C) [67].

### 3.2. Risks to Aquatic Organisms

In evaluating genotoxic risks to aquatic organisms, *Oreochromis niloticus* (Tilapian fish) were exposed to recombinant Bt spore-crystals expressing Cry1Ia, Cry10Aa, and Cry1Ba6. Comet assay (single cell gel electrophoresis), micronucleus (MN) test, and nuclear abnormalities (NA) in peripheral erythrocytes did not induce MN or nuclear abnormalities, and DNA damage was observed only at the highest exposure level, resulting in low genotoxic risks [68]. Zebrafish (Danio rerio) were exposed to four endotoxins from spore-crystal cry1Aa, cry1Ab, cry1Ac, and cry2A from B. thuringiensis to explore adverse effects on their genome and embryos. Cry1Aa increased the micronucleus (MN) frequency in peripheral erythrocytes of adult D. rerio, while cry1Ab, cry1Ac, and cry2A did not show genotoxicity after 96 h of exposure at a concentration of 100 mg/L. Exposures to binary mixtures (cry1Aa + cry1Ac, 50:50 mg/L) and (cry1Aa + cry2A, 50:50 mg/L) for 96 h also showed significantly increased MN frequency. Other evaluated binary mixtures did not show genotoxicity. In the zebrafish embryo-larval study, all tested cry-endotoxins showed embryotoxicity and developmental delay after exposure to the concentrations of 25, 50, 100, and 150 mg/L for 96 h [69].

In another study, zebrafish were fed a diet based on Bt-maize for two generations. Growth and reproductive performance were not affected. Bt-maize containing the Cry1Ab endotoxin had no negative influence on intestine histomorphology; nor were any significant differences observed between generations [70].

A study was carried out with tadpoles of Xenopus laevis exposed to grains of transgenic rice (T1C-19) as well as non-transgenic rice grains (MH63). Exposures to Cry1Ca endotoxins occurred at 10 µg/L up to 100 µg/L. The authors evaluated many endpoints, such as metamorphosis completed, survival rate, body weight, body length, organ weight, and liver enzyme activity. There were no significant differences observed between tadpoles fed with T1C-19 and MH63 rice grains [71].

### 3.3. Risks to Mammals

A study by researchers from Germany and Denmark showed that transgenic rice expressing the Cry1Ab protein administered to rats for 90 days did not result in hematological disorders [72].

According to the USEPA [73], the Bt protein is not structurally related to any known food allergen or toxin and does not show any oral toxicity when administered at high doses.

A skin prick test with CryIA(b) pure protein was carried out in Portugal on human volunteers with positive histories of food allergy. None of the volunteers reacted to sensitization to this protein [74]. The in vitro digestibility test showed that cry-proteins are unstable in the presence of digestive fluids and are not persistent in the digestive system [62]. Mice orally exposed to different concentrations of spore-crystal Cry1Aa, Cry1Ab, Cry1Ac, or Cry2Aa showed increased leukocyte-type neutrophils and lymphocytes in the bloodstream, which could be a consequence of allergenic and inflammatory processes. Depending on the amount ingested, these cry-protoxins can not be completely broken down in the stomach and reach the intestine, where the pH becomes alkaline, activated to its toxic form and thus causing a reaction with mucosa [29].

Genetically modified corn based on Bt-cry toxins has been widely used to feed livestock. Cry 1Ab toxins were detected in the gastrointestinal tract of pigs. Cry 1Ab toxins were not totally degraded through the mid-gut of pigs, being detected by PCR and ELISA immunoassay [75]. However, another study reported that Cry1Ab toxins were significantly degraded by the bovine gastrointestinal tract. The observed results of ELISA assays revealed that only fragments with immunoactive epitopes of Cry1Ab reacted, producing a misinterpretation [76].

On the other hand, a Canadian study reported the presence of Cry1Ab protein circulating in the blood of pregnant women (0.19 ± 0.30 ng/mL) and the fetal cord (0.04 ± 0.04 ng/mL). This means that these proteins are not completely degraded by the gastrointestinal tract, thus reaching the peripheral blood, and they can cross the placenta [77]. In response to this study, Goldstein et al. [78] reinforce that detections may represent, at best, protein fragments, since, according to studies carried out and by the accepted intake of the Cry1Ab protein, it is expected that it is not biologically active after food processing. Mueller and Gorst [79] emphasize that several published studies show the presence of Bacillus thuringiensis and insecticide residues based on Bacillus thuringiensis in fruits and vegetables such as grapes, lettuce, and tomatoes, which are not GM crops, inferring that the possible presence of the protein would not necessarily be related to the consumption of GM food.

The narrow spectrum of activity of Bt crops has contributed to an increased abundance of some beneficial insects by reducing applications of conventional insecticides. A comparative risk assessment approach with non-transgenic crops resulted in huge reductions in insecticide application, showing benefits for the environment. More studies might be addressed to elucidate the uncertainties of the risks that these Cry toxins could cause to biological control agents, such as natural enemies [80,81].

## 4. New Trends with Microbiological Nano-Biopesticides

Nanotechnology has revolutionized the biopesticide field with nano-capsules, nanoparticles, and nanoliposomes for the target-oriented release of bioactive microbial agents. The development of nano-formulations based on Bt endotoxins offers several advantages, increasing efficacy at a low cost and in an eco-friendly way. The Cry proteins are loaded to nano-composites, maintaining their bioactivity while enhancing the stability and storage conditions [82].

Nano-composites of Bt with different bioactive Cry proteins have been produced as nano-capsules, nano-suspensions, and nano-emulsion. These nano-formulations release bioactive endotoxins with more precision, demonstrating a higher capacity of controlled-release behavior, ensuring their effectiveness in long-term use [83]. Due to their small size, large surface, high solubility, and versatility, microbiological nano-formulations have the potential to achieve a step forward in insect-pest management with lower risks to humans and the environment.

## 5. Concluding Remarks

This review does not exhaust the matter but makes it clear that most studies examining adverse effects on the environment or ecotoxicity to non-target species have reported low risks or low ecotoxicity.

It should be noted that comparing ecotoxicity studies can sometimes be very difficult because the concentration units used in the papers differ. Spores, CFU, ppm, conidia, occlusion bodies, PIB, and other concentration/dose units confuse and may often generate doubts regarding the regulatory or registration decisions.

Bacillus thuringiensis is the most widely applied microbial insecticide, including its endotoxins cloned in genetically modified crops, and it is, therefore, the most frequently studied and well-known biopesticide around the world.

Due to the contradictory data and uncertainties that are sometimes observed, more studies on the safety of food derived from transgenic plants should be required by governments, mainly related to human consumption and potential effects on human health.

## Figures and Tables

**Table 1 ijerph-19-16495-t001:** Aquatic and terrestrial ecotoxicity of bacteria-based pesticides.

Bacteria or Component	Tested Organisms	Endpoint (Adverse Effects)	Doses or Concentrations	Ref.
*Bti*	*Chirocephalus grubei * (Crustacea)	Mortality 48 h (57% and 100%)	18 and 180 ppm	[9]
*Bti*	*Daphnia similis* (Crustacea)	Mortality 48 h (Absence)	1.5 × 10^6^ CFU/mL	[10]
*Bti*	*Daphnia magna* (Crustacea)	Mortality (54% and 80%)	4000 and 5000 ppm	[12]
*Bti*	*Biomphalaria glabrata* (Gastropoda)	Mortality 30 days (LC50)	1.5 × 10^7^ CFU/mL	[10]
*Bti*	*Danio rerio*/*Oreochromis niloticus* (Fish)	Mortality 30 days (Absence)	5.0 × 10^6^ CFU/mL	[16]
*Bti*	*Pimephales promelas* (Fish)	Mortality 96 h (60 and 100%)	2.0 and 6.5 × 10^6^ CFU/mL	[17]
*Bti* commercial	*Hyphessobrycon eques* (Fish)	Mortality 30 days (Absence)	1.0 × 10^6^ CFU/mL	[21]
*Bti* commercial	*Procambarus clarkii* (Fish)	Mortality 96 h (LC50)	103 ppm	[25]
Cry1C protein	*Daphnia magna* (Crustacea)	Reproduction/Development 21 days (Absence)	500 µg/L	[11]
*Btk* commercial	*Oncorhynchus mykiss* (Fish)	Mortality 32 days (20%)	2.9 × 10^9^ CFU/L	[14]
*Bti*, *Btk* and Isolated toxins	*Trichostrongylus colubriformis* (Soil Nematode)	Embryolethality 24 h (LD50)	0.0001 to 130 µg/mL	[18]
*Btk*	*Digitonthophagus gazela* (Soil Insecta)	Mortality 30 days (LC50)	1.3 ×10^6^ CFU/g	[10]
*Bt* commercial	*Colinus virginianus* *Anas platyrhynchus* (Birds)	Mortality and Pathogenicity (5 days) (Absence)	4.0 × 10^9^ to 3.4 × 10^11^ CFU/kg/day	[19]
*Bs* two strains	*Daphnia similis* (Crustacea)	Mortality 24 h (LC50)	185 to 190 µg/mL	[24]
*Bs*	*Daphnia similis* (Crustacea)	Mortality 48 h (Absence)	1.5 × 10^6^ CFU/mL	[10]
*Bs* two strains	*Streptocephalus dichotomus* (Crustacea)	Mortality 24 h (LC50)	107 to 115 µg/mL	[24]
*Bs* two strains	*Tubifex tubifex* (Annelid)	Mortality 24 h (LC50)	175 µg/mL	[24]
*Bs*	*Biomphalaria glabrata* (Gastropoda)	Mortality 30 days (Absence)	5 × 10^7^ CFU/mL	[10]
*Bs* two strains	*Rana bufo* tadpoles 16 mg (Anfibia)	Mortality 24 h (LC50)	300 to 470 µg/mL	[24]
*Bs* commercial	*Procambarus clarkii* (Fish)	Mortality 96 h (LC50)	75 ppm	[25]
*Bs* commercial	*Hyphessobrycon eques* (Fish)	Mortality 30 days (Absence)	1.0 × 10^6^ CFU/mL	[21]
*Bs*	*Danio rerio*/*Oreochromis niloticus* (Fish)	Mortality 30 days (Absence)	5 × 10^6^ CFU/mL	[16]
*Bs*	*Digitonthophagus gazela* (Soil Insecta)	Mortality 30 days (Absence)	1.75 × 10^6^ CFU/mL	[10]

Abbreviations: *Bti—Bacillus thuringiensis israelensis*, *Btk—Bacillus thuringiensis kurstaki*, *Bs—Bacillus sphaericus*, LC50—Lethal Concentration to 50%, CFU—Colony Formers Unit, ppm—parts per million.

**Table 2 ijerph-19-16495-t002:** Aquatic and terrestrial ecotoxicity of fungi-based pesticides.

Fungi	Tested Organisms	Endpoint (Adverse Effects)	Doses or Concentrations	Ref.
*Lagenidium giganteum*	*Ceriodaphnia dubia* (Crustacea)	Mortality 48 h/96 h (LC50)	8200 and 6700 Zoospores/mL	[33]
*Lagenidium giganteum*	*Daphnia pulex* (Crustacea)	Mortality 48 h/96 h (LC50)	7700 and 7700 Zoospores/mL	[33]
*Lagenidium giganteum*	*Daphnia magna* (Crustacea)	Mortality 48 h/96 h (LC50)	11,200 and 9400 Zoospores/mL	[33]
*Lagenidium giganteum*	*Chironomus tentans* (Insecta)	Mortality 96 h (LC50)	>50,000 Zoospores/mL	[33]
*Trichoderma stromaticum*	*Selenastrum capricornutum* (Algae)	Growth rate 168 h (Absence)	10^6^ spores/mL	[34]
*Trichoderma stromaticum*	*Daphnia similis* (Crustacea)	Reproduction 21 days (30%)	10^6^ spores/mL	[34]
*Trichoderma stromaticum*	*Hyphessobrycon scholzei* (Fish)	Mortality 30 days (Absence)	10^6^ spores/mL	[34]
*Metarhizium anisopliae*	*Menidia beryllina* embryos (Fish)	Malformations 9 days (50%)	10^6^ spores/mL	[35]
*Metarhizium anisopliae*	*Mysidopsis bahia* (Crustacea)	Mortality 96 h (LC50)	2.41 mg/L	[36]
*Metarhizium anisopliae*	*Palaemonetes pugio* embryos (Crustacea)	Mortality 15 days (LC50)	10^5^ spores/mL	[36]
*Metarhizium anisopliae*	*Xenopus laevis* larvae (Amphibia)	Mortality 96 h (LC50)	31.5 mg/L	[36]
*Metarhizium anisopliae*	*Gambusia affinis* (Fish)	Mortality 96 h	141 mg/L	[36]
*Metarhizium anisopliae*	*Melanotaenia duboulayi* (Fish)	Mortality 8 days (Absence)	2.0 × 10^6^ conidia/mL	[37]
*Metarhizium anisopliae*	*Ceriodaphnia dubia* (Crustacea)	Mortality 48 h (100%)	2.0 × 10^6^ conidia/mL	[40]
*Metarhizium anisopliae*	*Rana pipiens* (Amphibia)	Mortality via Oral (Absence)	1.0 × 10^9^ spores	[42]
*Beauveria bassiana*	*Mysidopsis bahia* (Crustacea)	Mortality 96 h (45%)	1.5 × 10^6^ spores/mL	[38]
*Beauveria bassiana* commercial	*Chironomus riparius* (Insecta)	Mortality 48 h (LC50)	34.7 mg/L	[39]
*Beauveria bassiana*	*Menidia beryllina* embryos/larvae (Fish)	Malformations 9 days (Presence)	1.0 × 10^6^ spores/mL	[35]
*Sporothrix insectorum*	*Biomphalaria glabrata* (Gastropoda)	Mortality 30 days (Absence)	5.0 × 10^7^ spores/mL	[10]
*Sporothrix insectorum*	*Digitonthophagus gazela* (Soil Insecta)	Mortality 30 days (Absence)	1.25 × 10^7^ spores/mL	[10]
*Colletotrichum gloeosporioides*	*Palaemonetes pugio* embryos/larvae (Crustacea)	Mortality 13 days (Absence)	1 × 10^6^ spores/mL	[44]

Abbreviations: LC50—Lethal Concentration to 50%.

**Table 3 ijerph-19-16495-t003:** Results of ecotoxicity assays of the Nucleopolyhedrovirus *Lymantria dispar* and *Orgyia pseudotsugata* on non-target species.

Assay	Type of Study	Results	Reference
Freshwater Invertebrates Test	Lethality of *Daphnia magna*, *Notonecta undulata*, and *Chironomus thummi*	LC_50_ > 250 PIB/mL	[51]
Freshwater Fish Toxicity/Pathogenicity Test	96 h Lethality of Brown trout and Blue gill sunfish	LC_50_ > 1.5 × 10^9^ PIB/gram	[52]
Honey Bee Toxicity/Pathogenicity Test	4 months feeding study	Absence of effects on egg laying, brood rearing, and honey production 10,850 AU_GL_/individual	[53]
Avian Oral Acute Toxicity/Pathogenicity Test	Feeding study with bobwhite quail	No signs of toxicity or pathogenicity in doses of 3.73 × 10^3^ PIB/g/individual	[54]
	8-day dietary feeding study mallard	LC50 > 16000 ppm	[51]
	Feeding study with birds	The birds were fed larvae infected with 3 × 10^7^ to 2 × 10^8^ PIB, and no effects were observed	[51]
	English sparrow	LD50 > 1969 mg/kg	[55]
Wildlife Mammalian Toxicity/Pathogenicity Test	Feeding study with mouse, short-tailed shrew, and Virginia opossum	The mammals were fed larvae infected with 4 × 10^8^ to 6 × 10^8^ PIB, and no short-term effects were found	[51]

Abbreviations: LC50—Lethal Concentration to 50%, PIB—polyhedral inclusion bodies, AU_GL_—activity unit of insect strain GL-1.

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
