# Peer review of "The Ecotoxicology of Microbial Insecticides and Their Toxins in Genetically Modified Crops: An Overview"

_ijerph, 2022, doi:10.3390/ijerph192416495_

Round 1

Reviewer 1 Report

The authors of the review manuscript “The ecotoxicology of microbial insecticides and their toxins in genetically modified crops: an overview” present a solid work on compilation of literature and a critical analysis of it, giving thorough insight on the role that Cry proteins have in toxic effects on modified crops. In addition, it is a pleasant and clear reading. Therefore, I would recommend it for publication after some minor changes as proposed below.

Line 33-36: I suggest that the authors make an introductory statement about Bacillus thuringiensis, prior to pointing out the importance and widespread use.

Line 37-38: This sentence is a bit confusing, what do the authors want to convey?

Line 39-40: Suggestion to rewrite the sentence as follows:

“The hazard potential is not based only on the toxic effects of compounds, but also on their uptake and elimination kinetics, their bioavailability, dispersion or accumulation in the environment”

Line 45: Considering the focus on environment, does it make sense to include “toxicological” on top of “ecotoxicological” assays?

Line 54: What is considered a “conventional toxicological test”?

Line 59: Remove “thus”.

Line 79: Replace “were” by “have been”.

Line 95-100: Suggestion to move this paragraph further down, prior to the effects of Bt in birds (line 115), that way it is organized firstly by aquatic and then by terrestrial ecosystems.

Section 2 and 3: I suggest that the authors organize these by sub-sections similar to section 5, for soil and aquatic organisms.

Table 1: Apart from the bacteria strain-based organization, the information seems a bit all over the place. I suggest that the authors alternate a white and grey background for different bacteria strains, in order to distinguish when one starts and the other ends.

Table 3: There is some incoherence in the presentation of species in the table, as well as the absence of the original reference of the study. Please revise this.

Line 333: There is the lack of a reference supporting the affirmation on the studies performed in earthworms, springtails and so on.

Line 357-360: Please present first the results obtained using the assay, then affirm that there is low genotoxic risk.

Author Response

Thank you for the review.

Obs. The lines number can be changed with the re-organization of the text.

REVIEWER 1  

The authors of the review manuscript “The ecotoxicology of microbial insecticides and their toxins in genetically modified crops: an overview” present a solid work on compilation of literature and a critical analysis of it, giving thorough insight on the role that Cry proteins have in toxic effects on modified crops. In addition, it is a pleasant and clear reading. Therefore, I would recommend it for publication after some minor changes as proposed below.

Line 33-36: I suggest that the authors make an introductory statement about Bacillus thuringiensis, prior to pointing out the importance and widespread use.

R. The sentence has been modified to improve the understanding of the text. Bacillus thuringiensis will be better described in item 2.

Line 37-38: This sentence is a bit confusing, what do the authors want to convey?

R. The second paragraph of the introduction presents ecotoxicology as the science responsible for studying the effects of “chemical” agents on ecosystems. However, because several of the MPCAs have their mode of action through “toxins”, their effects, transformations and environmental kinetics must be studied by this science.

Line 39-40: Suggestion to rewrite the sentence as follows:

“The hazard potential is not based only on the toxic effects of compounds, but also on their uptake and elimination kinetics, their bioavailability, dispersion or accumulation in the environment”

R. Changed as suggested.

Line 45: Considering the focus on environment, does it make sense to include “toxicological” on top of “ecotoxicological” assays?

R. In our understanding, and as described in the text, for health registration purposes, toxicological studies are focused on mammals with the aim of protecting human health. On the other hand, ecotoxicological studies are main focused on generating data aiming to protect other living species present in the environment. This is a matter of terminology and conceptualization.

Line 54: What is considered a “conventional toxicological test”?

R. The term “conventional” has been changed to “classic” and it means the standard toxicological test where the effects of a chemical substance are evaluated on a population based on its exposure to several doses or concentrations.

Line 59: Remove “thus”.

R. Changed as suggested.

Line 79: Replace “were” by “have been”.

R. Changed as suggested

Line 95-100: Suggestion to move this paragraph further down, prior to the effects of Bt in birds (line 115), that way it is organized firstly by aquatic and then by terrestrial ecosystems.

R. Changed as suggested.

Section 2 and 3: I suggest that the authors organize these by sub-sections similar to section 5, for soil and aquatic organisms.

R. Topics were re-organized by sub-sections as suggested.

Table 1: Apart from the bacteria strain-based organization, the information seems a bit all over the place. I suggest that the authors alternate a white and grey background for different bacteria strains, in order to distinguish when one starts and the other ends.

R. Table 1 was better organized by alternating the colours of the lines according to the strains tested as suggested by reviewer 1. The same method was used to Table 2.

Table 3: There is some incoherence in the presentation of species in the table, as well as the absence of the original reference of the study. Please revise this.

R. In table 3, the species tested are presented in evolutionary order, including the presence of pathogenicity assays. The data presented in table 3 were fully taken from the registration document of the viruses published by USEPA. Most of the references cited in the USEPA document, are unpublished reports not available on-line, even so all references presented in the document were included in the manuscript.

Line 333: There is the lack of a reference supporting the affirmation on the studies performed in earthworms, springtails and so on.

R. The citation was inserted after the referred text.

Line 357-360: Please present first the results obtained using the assay, then affirm that there is low genotoxic risk.

R. Changed as suggested

Reviewer 2 Report

I understand and the authors mention it in the abstract but the diversity of units used make things difficult to compare. In fact it make reading the comparison also most impossible. Could the author consider converting some of the units to one standard if possible? In a table maybe. It would really add to the ease of comparison and hence understand. Indeed to the novelty of the paper

L 27 “ agriculture and in the control of mosquitoes.”  Reference needed

L32  “Or if the quantity is safe for environmental

L233 “at” the concentration of 2.0 x 106 conidia/mL

Table 3 is there no concentration information associated with the virus studies?

L320 “requiring”

L 345 “The persistence of these cry toxins in soils depends on  microbiological activity, solubilization, soil pH, adsorption by soil organominerals, temperature and UV radiation.”

Is there any more information on this above sentence? For instance what pHs effects the persistence.

A brief sentence is given that high temperature and uv intensity. How the half-life influenced by these outlined factors.

Microbiological particles could have a massive impact on the air quality has there been any work done on the concentrations introduced into the atmosphere or long range of these particles. I feel this should be mentioned in the New trends with microbiological nano-biopesticides Section

Author Response

Thank you for the review. The lines number can be changed after the re-organization of the text.

REVIEWER 2

I understand and the authors mention it in the abstract but the diversity of units used make things difficult to compare. In fact it make reading the comparison also most impossible. Could the author consider converting some of the units to one standard if possible? In a table maybe. It would really add to the ease of comparison and hence understand. Indeed to the novelty of the paper.

R. The present review used the data fully as presented by the collected publications. In fact, it would be great to be able to do the conversion to present the information in a more comparable way. However, published data do not provide sufficient detail on doses or concentrations to accomplish this task.

For example, data presented in ppm or ppb, mg/L or µg/mL can be compared with each other, since they present a relation between a weight and a volume. However, others units like spores per kilogram or conidia per gram or spores per milliliter or occlusion bodies or PIB, there is no way to compare as there is no relationship with a weight.

Thus, the authors preferred to maintain the units originally presented in the articles used, reinforcing the difficulty in comparison and the need for standardization of measurement units between different tests.

L 27 “ agriculture and in the control of mosquitoes.”  Reference needed

R. The citation is found at the end of the second sentence [1].

L32  “Or if the quantity is safe for environmental”

R. Changed as suggested

L233 “at” the concentration of 2.0 x 106 conidia/mL

R. Changed as suggested

Table 3 is there no concentration information associated with the virus studies?

R. Table 3 was presented in the same way as it is in the USEPA document. Concentration or dose information and observed effect are shown in the Results column. 

 L320 “requiring”

R. Changed as suggested

L 345 “The persistence of these cry toxins in soils depends on microbiological activity, solubilization, soil pH, adsorption by soil organominerals, temperature and UV radiation.”

Is there any more information on this above sentence? For instance what pHs effects the persistence.

A brief sentence is given that high temperature and uv intensity. How the half-life influenced by these outlined factors.

R. The text was modified presenting what was researched and the main results obtained related to half-life and temperature.

Microbiological particles could have a massive impact on the air quality has there been any work done on the concentrations introduced into the atmosphere or long range of these particles. I feel this should be mentioned in the New trends with microbiological nano-biopesticides Section.

R. In fact, there are still few studies involving to nanobiopesticides and mainly related to the adverse effects of it. In our perpective, regarding ecotoxicological concern, object of this review, the toxicological information presented here, can also predict the potential effects of encapsulated bioproducts, since their main functions would be to increase the environmental persistence, but with the same active ingredients, the microbial pesticides.

Reviewer 3 Report

With an increase in concern about environmental protection and climate change, the authors did a good work in compiling this review on ecotoxicology of microbial insecticides. The information put together sounds scientific with a good overall writing style.  

Although one will conclude microbial insecticides, products and toxins tested present low toxicity and low risk, when compared to the concentrations used for pesticide purposes; these still raise eyebrows for complementary studies for possible effects on human health.

I will recommend the authors to consider my comments and suggestions provided below in two sections as general and specific comments.

General comments:

1.     Table 1 and 2-foot note should contain all necessary abbreviations and measurement units explained. This will let the table standalone. Let the title of the tables (1 and 2) be explicit enough.

2.     References should be consistent. Please check out references 2 and 63.

Specific comments:

1.     Line 11: I propose deleting the words “and their”. This keeps the sentence sense and makes it clearer.

2.     Line 17-18: “The data presented show that many results are difficult to compare, due to the diversity of measurement units used in the papers”. Comment: it is hard for the reader to understand which of the results are difficult to compare…within or between articles?

3.     Line 25: delete “,” between shown and over

4.     Line 94: 30-day…30-days?

5.     Line 216: change “report” to reported

6.     Line 227: delete “In this same study”, start a sentence with…The authors….

7.     Line 230: delete one of the word “found”.

8.     Line 303: 14-days OR 14-day? I am not sure.

9.     Line 345, 363-366: change “cry” to Cry

10.  Line 351: Delete “condition of”.

11.  Line 355: In italics (Oreochromis niloticus)

12.  Line 390: “in vitro

13.  Line 402, 406 - 412: check out the right way of writing Cry 1Ab…consistency.

14.  Line 414 and 415: “Bacillus thuringiensis”

Author Response

Thank you for the review. The lines number can be changed after the re-organization of the text.

REVIEWER 3

With an increase in concern about environmental protection and climate change, the authors did a good work in compiling this review on ecotoxicology of microbial insecticides. The information put together sounds scientific with a good overall writing style.  

Although one will conclude microbial insecticides, products and toxins tested present low toxicity and low risk, when compared to the concentrations used for pesticide purposes; these still raise eyebrows for complementary studies for possible effects on human health.

I will recommend the authors to consider my comments and suggestions provided below in two sections as general and specific comments.

General comments:

1. Table 1 and 2-foot note should contain all necessary abbreviations and measurement units explained. This will let the table standalone. Let the title of the tables (1 and 2) be explicit enough.

R. All abbreviations were included in tables 1, 2 and 3. The titles of tables 1 and 2 were better described.

2. References should be consistent. Please check out references 2 and 63.

R. Sorry to reviewer, but we do not understand the comment. References 2 and 63 are consistent and complete the manuscript according with information the authors intend to present.

Specific comments:

1. Line 11: I propose deleting the words “and their”. This keeps the sentence sense and makes it clearer.

R. Thank you for your note, but in the case of the sentence in question we are talking about microbial insecticides and their toxins separately and not just the toxins. So we disagree with the reviewer.

2. Line 17-18: “The data presented show that many results are difficult to compare, due to the diversity of measurement units used in the papers”. Comment: it is hard for the reader to understand which of the results are difficult to compare…within or between articles?

R. It was included in the sentence that the difficulty is due to the diversity of measurement units used by the different research data, that is, comparison between articles.

3. Line 25: delete “,” between shown and over

R. Changed as suggested.

4. Line 94: 30-day…30-days?

R. 30 days. Changed as suggested.

5. Line 216: change “report” to reported

R. Changed as suggested.

6. Line 227: delete “In this same study”, start a sentence with…The authors….

R. Changed as suggested.

7. Line 230: delete one of the word “found”.

R. Changed as suggested.

8. Line 303: 14-days OR 14-day? I am not sure.

R. 14 days. Changed as suggested.

9. Line 345, 363-366: change “cry” to Cry

R. Changed as suggested.

10. Line 351: Delete “condition of”.

R. The sentence was modified as requested by reviewer 2.

11. Line 355: In italics (Oreochromis niloticus)

R. Changed as suggested.

12. Line 390: “in vitro

R. Changed as suggested.

13. Line 402, 406 - 412: check out the right way of writing Cry 1Ab…consistency.

R. All forms were written the same way as suggested.

14. Line 414 and 415: “Bacillus thuringiensis”

R. Corrected as suggested.